# Ketone Monoester Followed by Carbohydrate Ingestion after Glycogen-Lowering Exercise Does Not Improve Subsequent Endurance Cycle Time Trial Performance

**DOI:** 10.3390/nu16070932

**Published:** 2024-03-23

**Authors:** Manuel D. Quinones, Kyle Weiman, Peter W. R. Lemon

**Affiliations:** Exercise Nutrition Research Laboratory, School of Kinesiology, The University of Western Ontario, London, ON N6A 3K7, Canada; mquinon2@uwo.ca (M.D.Q.); kweiman4@uwo.ca (K.W.)

**Keywords:** exogenous ketones, ketosis, glycogen repletion, exercise recovery, insulin

## Abstract

Relative to carbohydrate (CHO) alone, exogenous ketones followed by CHO supplementation during recovery from glycogen-lowering exercise have been shown to increase muscle glycogen resynthesis. However, whether this strategy improves subsequent exercise performance is unknown. The objective of this study was to assess the efficacy of ketone monoester (KME) followed by CHO ingestion after glycogen-lowering exercise on subsequent 20 km (TT_20km_) and 5 km (TT_5km_) best-effort time trials. Nine recreationally active men (175.6 ± 5.3 cm, 72.9 ± 7.7 kg, 28 ± 5 y, 12.2 ± 3.2% body fat, VO_2_max = 56.2 ± 5.8 mL· kg BM^−1^·min^−1^; mean ± SD) completed a glycogen-lowering exercise session, followed by 4 h of recovery and subsequent TT_20km_ and TT_5km_. During the first 2 h of recovery, participants ingested either KME (25 g) followed by CHO at a rate of 1.2 g·kg^−1^·h^−1^ (KME + CHO) or an iso-energetic placebo (dextrose) followed by CHO (PLAC + CHO). Blood metabolites during recovery and performance during the subsequent two-time trials were measured. In comparison to PLAC + CHO, KME + CHO displayed greater (*p* < 0.05) blood beta-hydroxybutyrate concentration during the first 2 h, lower (*p* < 0.05) blood glucose concentrations at 30 and 60 min, as well as greater (*p* < 0.05) blood insulin concentration 2 h following ingestion. However, no treatment differences (*p* > 0.05) in power output nor time to complete either time trial were observed vs. PLAC + CHO. These data indicate that the metabolic changes induced by KME + CHO ingestion following glycogen-lowering exercise are insufficient to enhance subsequent endurance time trial performance.

## 1. Introduction

Carbohydrate (CHO) is considered an important exercise fuel because fatigue during prolonged exercise at moderate to high intensities is associated with glycogen depletion [1,2,3], which occurs typically within ~1–2 h with continuous exercise [3,4]. Further, greater initial muscle glycogen content has been shown to correlate with better performance in both intense, intermittent sports [4,5] and prolonged endurance efforts [6]. As such, repletion of CHO stores immediately after exercise is an important factor that determines performance in subsequent events, especially if these occur within 24 h [2]. Dose-response studies have determined that CHO ingestion of ~1.2 g·kg^−1^·h^−1^ is the optimal acute recovery dose to optimize glycogen repletion; i.e., there is no apparent benefit with greater doses [2]. Protein in combination with CHO has also been proposed to enhance glycogen resynthesis when 1 part protein is provided with ~4 parts CHO [7,8]. While both strategies have resulted in better exercise performance a few hours after glycogen-lowering exercise [2,7,8], it is important to note that the addition of protein may be beneficial for glycogen replenishment only if CHO ingestion during recovery is suboptimal (<0.8 g·kg^−1^·h^−1^) [9].

Ketone bodies (KBs) are water-soluble, lipid-derived metabolites produced in the liver during periods of low CHO availability such as starvation, prolonged exercise, uncontrolled diabetes or dietary manipulations [10,11]. Once synthesized, they are released into the bloodstream, primarily in the form of beta-hydroxybutyrate (βHB), to be transported to extrahepatic tissues where they can be oxidized. Tissues such as the brain, kidney, as well as both skeletal and cardiac muscle have been identified as sites of KB utilization [12]. Recently, exogenous ketone supplement ingestion has emerged as an effective way to elevate circulating KBs. Further, it has been proposed that they could serve as an alternative fuel during exercise, thereby sparing CHO [13], although some data suggest that the contribution of exogenous ketones to energy expenditure during exercise at intensities corresponding to 25, 50 and 75% of Wmax is very small [14]. This is particularly true in the presence of other substrates. KBs may also serve as a critical signaling metabolite during exercise recovery [15], meaning ingestion might enhance post-exercise protein synthesis by increasing mTORC1 activation [16]. This increased activation was explained by a decreased activation of AMPK, known to have an inhibitory effect on mTORC1. If so, ketone monoester (KME) ingestion could facilitate muscle repair after exercise. Alternatively, KME could be beneficial due to incretin effects [17], i.e., a greater insulinotropic response resulting in enhanced muscle glycogen resynthesis. Unfortunately, research in this area is scarce and controversial. In animal studies, exogenous KBs do enhance glycogen content when combined with CHO [18,19,20]. In humans, Holdsworth et al. [21] found that glycogen repletion was 50% greater when KME was ingested vs. a placebo prior to a 2 h CHO infusion post-exercise. However, the hyperglycemic clamp used to maintain glucose concentrations at ~10 mM throughout recovery is impractical for athletes. The only other study that examined this question in humans was conducted by Vandoorne et al. [16]. In their study, participants consumed a high dose protein-CHO solution containing 1.0 g·kg^−1^·h^−1^ CHO and 0.3 g·kg^−1^·h^−1^ of protein together with either KME or placebo for 5 h following glycogen-depleting exercise. No differences in glycogen repletion were observed but the KME group showed increased activation of mTORC1 and this might benefit exercise performance via enhanced post-exercise muscle protein synthesis, as mentioned above. Unfortunately, exercise performance was not assessed in these studies, so it is unclear whether the ingestion of KME followed by CHO after glycogen-depleting exercise can enhance subsequent exercise performance. Therefore, the purpose of the present study was to assess the efficacy of KME + CHO ingestion following glycogen-lowering exercise on subsequent 20 km (TT_20km_) and 5 km (TT_5km_) best-effort time trials. We hypothesized that post-exercise KME + CHO ingestion would improve TT performance.

## 2. Materials and Methods

### 2.1. Participants

Nine men were recruited for this study (Table 1). All were involved in some type of endurance training at least 3 days per week for 6 months or more prior to the study. Each completed both a physical activity readiness questionnaire (PAR-Q+) [22] and a health information form to minimize any potential contraindications to exercise. All potential risks were explained fully prior to any testing, and the participants provided written, informed consent for the study protocol approved previously by the Western University Office of Research Ethics (project ID: 113713).

### 2.2. Study Design

A double-blind, randomized, crossover research design was implemented involving two preliminary and two experimental sessions. The preliminary sessions were conducted before any experimental sessions began to collect baseline data as well as to familiarize participants with all the procedures to be implemented during the experimental sessions. These preliminary sessions were separated by at least two days and the second session was completed at least five days before starting the experiment. The two experimental sessions, separated by at least one week, were conducted at the same time of the day, and were rotated systematically to prevent order effects. Further, to minimize food intake differences across treatments, participants recorded their entire food/drink intake for the 2 days preceding the first experimental session and replicated this intake for their second trial. Experimental sessions were comprised of a glycogen-lowering exercise protocol followed by a 4 h recovery period where participants received supplementation drinks (see below for details). Subsequently, participants performed TT_20km_ and TT_5km_ best-effort time trials to assess the impact of this recovery nutrition on endurance performance (Figure 1).

### 2.3. Preliminary Sessions

During the initial familiarization visit, a body composition test (Bod Pod^®^, COSMED, Concord, CA, USA) for percent body fat, as described previously [23], and an incremental ramp protocol to volitional fatigue on a Velotron cycle ergometer (Racer Mate, Seattle, WA, USA) for peak cycle power output (Wmax) and VO_2_max were completed. Briefly, participants performed a 2 min warm up at 100W followed by an increase of 1W every 3 sec until volitional exhaustion. Expired gases were collected via a breath-by-breath collection system (Sensormedics Vmax 29, Yorba Linda, CA, USA), previously calibrated according to manufacturer’s guidelines using known gas volumes and composition. The greatest value achieved over a 20 s collection period was considered max whenever a plateau in VO_2_ occurred (<50% of the expected increase in oxygen uptake for the increased workload) or when two of the following three criterion measures were attained (±10 bpm of age-predicted maximum HR, RER > 1.15 [RER = volume of CO_2_ produced/volume of O_2_ consumed] or volitional exhaustion). Heart rate (HR) was monitored throughout the test (Polar RST200^TM^, Polar Electro Inc., Lachine, QC, Canada).

At least 48 h following the first visit, participants visited the laboratory a second time to familiarize themselves with the TT_20km_ and TT_5km_ in order to minimize potential learning effects during the experiment.

### 2.4. Experimental Sessions

Participants reported to the laboratory at 0800 h following a 12 h overnight fast, with limited activity (drive/use of the elevator to get to the laboratory) and having abstained from strenuous exercise, caffeine or alcohol consumption for 24 h. Upon arrival, and after verbal confirmation of a fasted state, participants performed a glycogen-lowering exercise bout on a Velotron cycle ergometer. Briefly, the protocol began with a 10 min warm-up at a workload of 50% Wmax. Thereafter, participants engaged in an intermittent exercise protocol that involved cycling in one min intervals alternating between workloads of 90% (high-intensity effort) and 50% (recovery) of Wmax, respectively. When participants were fatigued (inability to maintain cadence at 60 pedal rev/min during the high-intensity effort), the workload was dropped to 80% Wmax. Subsequent reductions in workload to 70%, 60%, etc., were carried out progressively until participants could not complete a workload of 60% Wmax with a cadence of 60 rev/min. The recovery workload remained at 50% Wmax throughout. This protocol was adapted from previous studies [8,24]. The duration of the protocol was 56 ± 15.6 min. During the glycogen-lowering exercise, water was provided ad libitum and the amount ingested was recorded (640.4 ± 237.2 mL) and replicated in the subsequent experimental trial.

Following the glycogen-lowering exercise, CHO recovery drinks and the respective treatment were provided during the first 2 h of recovery. Participants ingested either a KME drink (TdeltaS Ltd., Thame Oxfordshire, UK) or isoenergetic CHO alone (PLAC-dextrose), followed by frequent CHO feedings (Dextrose monohydrate). The KME supplement contained 25 g of ketone esters, providing a dose of 347 ± 40.5 mg·kg BM^−1^. Capillary blood samples to measure glucose and βHB (FreeStyle Precision Neo^®^, Abbott Diabetes Care Limited, Witney, UK) were obtained immediately after the glycogen-lowering exercise and every 30 min during the first 2 h of recovery via the fingerprick method (Figure 1). Venous blood samples (~5 mL) to measure serum insulin concentration were collected from an antecubital vein immediately after the glycogen-lowering exercise and each hour for the first 2 h of recovery, using a 21-gauge vacutainer needle and serum separator tubes containing a clot activator. These samples were allowed to clot on ice for 30 min and then centrifuged at 3000 rpm for 10 min at 4 °C (Allegra 21R; Beckman Coulter, Mississauga, ON, Canada). The serum was collected and stored at −80 °C until analysis. Enzyme-linked immunosorbent assays (ELISAs) specific to human insulin (Crystal Chem Inc, ELK Grove Village, IL, USA) were performed on the serum in duplicate using a Biotek Synergy H1 microplate reader (Santa Clara, CA, USA).

Participants received either KME or an isoenergetic appearance-matched placebo (dextrose) after the initial blood sample. Thereafter, recovery drinks containing only CHO were supplied every 15 min for the initial 2 h of the 4 h recovery period (a 10–12% solution at a rate of 1.2 g·kg^−1^·h^−1^). To avoid any potential GI distress during performance tests, supplementation ended at 2 h. After the feeding period, participants remained in the laboratory for another 2 h (passive recovery) before engaging in sequential 20 and 5 km best-effort time trials (TT_20km_ and TT_5km_) (Figure 1).

### 2.5. Time Trials (TT_20km_ and TT_5km_)

When the 4 h recovery period following the glycogen-lowering exercise ended, participants performed a 10 min warm-up cycling at 100 W on a Velotron cycle ergometer, a best-effort TT_20km_, a 5 min recovery period and a best-effort TT_5km_. Two trials were used to minimize pacing strategies and to ensure the intensity was high enough to challenge glycogen stores. During both TTs, participants received only distance traveled feedback. Average power output and time to finish were recorded. Water was available ad libitum throughout the time trial and the intake was recorded (750.5 ± 502.3 mL) so it could be reproduced during the subsequent treatment.

### 2.6. Statistical Analysis

Statistical analyses were performed using SigmaPlot for Windows (Version 12.5, SYSTAT, San Jose, CA, USA). Blood metabolite concentrations were analyzed using two-way (condition by time) repeated-measures ANOVA. Post hoc Tukey’s honest significant difference testing was used, where necessary. All variables from the TT_20km_ and TT_5km_ were analyzed using paired *t*-tests. Effect sizes (Cohen’s d) were calculated for all blood and performance variables and interpreted as small (0.2), medium (0.5) and large (0.8). Significance was set at *p* ≤ 0.05. Data are presented as means ± SD.

## 3. Results

### 3.1. Blood Ketones (Beta-Hydroxybutyrate—βHB)

There was a significant interaction effect (*p* < 0.001) for blood beta-hydroxybutyrate (βHB) concentration. Pairwise comparisons indicated that blood βHB was significantly greater for the KME + CHO group at 30 (2.7 ± 0.7 vs. 0.1 ± 0.1 mM, *p* < 0.001; *d* = −5.2), 60 (2.6 ± 0.9 vs. 0.1 ± 0.1 mM, *p* < 0.001; *d* = −3.9), 90 (1.9 ± 0.8 vs. 0.1 ± 0.0 mM, *p* < 0.001; *d* = −3.1) and 120 (1.2 ± 0.7 vs. 0.0 ± 0.1 mM, *p* < 0.001; *d* = −2.4) min post-treatment ingestion, compared to PLAC + CHO group (Figure 2).

### 3.2. Blood Glucose

A significant time × treatment interaction (*p* = 0.03) for blood glucose concentration was seen (Figure 3). Pairwise comparisons showed that blood glucose was significantly lower at 30 min (5.8 ± 0.6 vs. 7.1 ± 0.8 mM, *p* = 0.003; *d* = 1.8) and 60 min (6.1 ± 1.2 vs. 7.2 ± 2.3 mM, *p* = 0.01; *d* = 0.6) for KME + CHO compared to PLAC + CHO.

### 3.3. Blood Insulin

There was a significant time × treatment interaction (*p* = 0.006) for blood insulin concentration (Figure 4). Pairwise comparisons indicated that blood insulin concentration was significantly greater at 2 h (28.1 ± 13.1 vs. 21.3 ± 11.8 mU·L^−1^, *p* = 0.002; *d* = −0.5) for KME + CHO compared to PLAC + CHO.

### 3.4. Measures of Exercise Performance

#### Time Trials (TT_20km_ and TT_5km_)

There were no significant differences between conditions in time to complete the best-effort TT_20km_ (38.86 ± 1.56 vs. 38.38 ± 2.25 min; *p* = 0.420; *d* = 0.2) (Figure 5A) nor the best-effort TT_5km_ (9.46 ± 0.88 vs. 9.48 ± 0.65 min; *p* = 0.884; *d* = −0.03) (Figure 5C). Likewise, there were no significant differences in average TT_20km_ power output (169.3 ± 25.6 vs. 174.1 ± 21.9 W; *p* = 0.427; *d* = −0.2) (Figure 5B) nor average TT_5km_ power output (186.3 ± 40.5 vs. 182.4 ± 31.8 W; *p* = 0.524; *d* = 0.1) (Figure 5D) between PLAC + CHO vs. KME + CHO treatments.

## 4. Discussion

The present study aimed to determine if KME followed by CHO ingestion during recovery from glycogen-lowering exercise could enhance subsequent endurance time trial performance compared to isoenergetic CHO supplementation alone. Our results indicate that, following glycogen-lowering exercise, KME + CHO ingestion displayed greater blood βHB, lower blood glucose during the first hour and greater insulinemia in the second hour compared to the PLAC + CHO treatment; however, there were no significant differences in any of the performance measures during the TT_20km_ or the TT_5km_, initiated 4 h after glycogen-lowering exercise. These data suggest that KME + CHO ingestion following glycogen-lowering exercise causes some metabolic changes compared to PLAC + CHO; however, endurance time trial performance is not enhanced.

These observations add to our understanding of the effects of KME ingestion on exercise recovery and subsequent endurance performance because limited data exist on the role of acute exogenous KME supplementation during post-exercise recovery in humans [16,21]. For instance, Holdsworth et al. [21] reported that a single dose of 573 mg·kg^−1^ KME followed by glucose infusion over 2 h following glycogen-lowering exercise resulted in greater muscle glycogen synthesis (50% greater vs. glucose alone). This would be expected to enhance subsequent endurance performance, but no performance measures were made. In addition, Vandoorne et al. [16] compared KME and a CHO-protein drink vs. a CHO-protein drink alone and found an initial oral dose of 500 mg·kg^−1^ KME and four subsequent doses of 250 mg·kg^−1^ KME at 1 h intervals for 5 h of recovery enhanced mTORC1 activation in human skeletal muscle after strenuous exercise, but with no differences in muscle glycogen synthesis. In our study, participants received a single dose of ~350 mg·kg^−1^ KME after glycogen-lowering exercise, and greater blood insulin concentrations were observed at 2 h (Figure 4) with KME ingestion (28 mU·L^−1^ vs. 21 mU·L^−1^) compared to the dextrose alone trial. Ingestion of βHB can exert an incretin effect that augments insulin secretion, compared to infusion of βHB [17], and this could explain our observations. However, while insulin secretion was greater at 2 h in our KME group, the lack of a difference in endurance time trial performance between our two conditions suggests glycogen availability may have been similar in both treatments. Holdsworth et al. [21] attributed the 50% increase in glycogen resynthesis with KME and CHO infusion to the difference in circulating insulin (31 mU·L^−1^ with KME + CHO vs. 16 mU·L^−1^ with CHO alone). Conversely, Vandoorne et al. [16] found no difference in glycogen content after recovery with no significant differences in circulating insulin (23 mU·L^−1^ with KME vs. 27 mU·L^−1^ in the comparison group). Perhaps our insulin differences were too small to affect glycogen storage.

Although it could be argued that differing glycogen stores would not necessarily produce endurance performance benefits, if they were sufficient with both treatments, we do not think this is likely in our study. In fact, we separated our endurance task into two-time trials (20 km followed by 5 km) to minimize pacing strategies which we have noticed in previous studies. This is important because, even if treatment glycogen stores differed, pacing strategies could eliminate performance effects, especially in our TT_20km_ because of its prolonged duration. However, this would be much less likely during the subsequent TT_5km_. because of its shorter duration. Further, the observed greater PO in the TT_5km_ vs. TT_20km_ (Figure 5) suggests that pacing was unlikely the determining factor. Consequently, the observed similar performance results are more likely the result of similar glycogen availability between treatments. In contrast, KME might have suppressed CHO use during the performance tests leading to similar performance efforts. Therefore, a different mix of energy substrates and/or even a difference in local muscle signaling could have occurred. Unfortunately, we do not have muscle glycogen measures so these arguments remain speculative.

Of course, differences in performance could also be related to glucose availability. It has been shown that βHB stimulates insulin release in the presence of 5 mM of glucose, at least in isolated rat pancreatic islets [25]. Alternatively, Holdsworth et al. [21] used a hyperglycemic clamp in vivo to maintain substantial blood glucose concentrations (~10 mM) throughout their experiments, which is not practical for athletes. In the present study, CHO ingestion occurred every 15 min during recovery at the highest recommended rate (1.2 g·kg^−1^·h^−1^) in an attempt to maximize glycogen resynthesis by ensuring a steady blood glucose concentration throughout recovery. However, blood glucose was lower (~6.1 mM) than that of Holdsworth et al. [21], suggesting that glucose availability may be insufficient without a glucose clamp, i.e., gastric emptying, of CHO may be a limiting factor. Consistent with this scenario is the Vandoorne et al. data [16], where no differences in glycogen content were observed with elevated blood βHB and a blood glucose concentration of ~6.5 mM. Glucose transport in the GI tract is the key factor in absorption and the capacity of the glucose transporter (sodium-dependent glucose transporter 1) requires ~1.3 to 1.7 g CHO·min^−1^ [26]. Importantly, these transporters can be upregulated with a chronic high CHO diet, i.e., ~8.5 g·kg^−1^·d^−1^ [27,28] but in our study, the participants consumed routinely modest amounts (3.5 g·kg^−1^·d^−1^) of CHO (Table 1) and this could have limited glucose availability during our feeding/recovery period. Clearly, more research is warranted to elucidate the possible benefits of KME supplementation during exercise recovery and investigators should ensure that both βHB concentration and glucose availability are sufficient.

Interestingly, in our study, the KME + CHO condition displayed lower glucose concentrations during the first hour of recovery (Figure 3). While it is possible that the observed differences in blood glucose were affected by the different amounts of CHO ingested initially and/or by insulin differences between groups, the reduced blood glucose observations are consistent with the glucose response seen by Vandoorne et al. [16] where blood glucose was also lower at 60 and 90 min of recovery. Moreover, this apparent hypoglycaemic action of βHB has also been reported during exercise after KME ingestion [13,29,30] as well as during βHB infusion studies in humans where the blood βHB concentration was set at ~2 mM [31,32]. Often these differences in circulating glucose are explained partly by hyperinsulinemia; however, this response can still occur even in the absence of high insulin, as observed in the Vandoorne et al. [16] study. So, this hypoglycemic response might be due to a decrease in hepatic glucose output (gluconeogenesis) because βHB infusion studies have shown a 20% decrease in endogenous glucose production in healthy males [31,32]. Unfortunately, these variables were not measured in the present study, so the details must await future studies. Nonetheless, the blood glucose-lowering action of βHB might have important therapeutic implications. For instance, recently Soto-Mota et al. [33] found that three times per day KME supplementation for 4 wk was safe, well tolerated and resulted in improved glycaemic control in type 2 diabetes patients.

Finally, based on our sample size calculations using data from Holdsworth et al. [21], the present study included sufficient participants, but the sample studied was still small. This has implications for the generalizability of our results because detecting smaller effects typical of exercise performance evaluations is difficult, i.e., Type 2 statistical errors are possible. Future studies with larger samples are recommended to confirm and extend the current findings. Further, although there is no question that our protocol would have lowered muscle glycogen, glycogen content was not quantified, so our results remain inconclusive. Incorporating these measures in future research studies would provide a more comprehensive understanding of the data. Also, we studied men only, so the responses in women remain unknown. Future experiments should aim to elucidate potential sex-specific effects in order to enhance the external validity of the findings.

## 5. Summary and Conclusions

In summary, KME + CHO ingestion following glycogen-lowering exercise reduced blood glucose during the first hour of the recovery period, increased blood βHB throughout 4 h of the recovery period, and increased blood insulin concentration 2 h into the recovery period vs. PLAC + CHO. However, no significant differences in any of the exercise performance parameters measured during the TT_20km_ and TT_5km_ were observed. These data suggest that the metabolic changes induced by the CHO + KME ingestion provided in this study are insufficient to elicit significant ergogenic benefits in subsequent endurance exercise performance trials. In the future, it is important to consider strategies that might improve both feeding protocols as well as CHO and KME doses to ensure adequate glucose and ketone availability.

## Figures and Tables

**Figure 1 nutrients-16-00932-f001:**
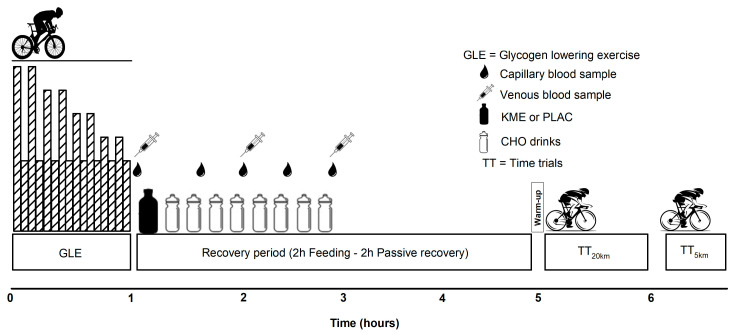
Overview of study protocol.

**Figure 2 nutrients-16-00932-f002:**
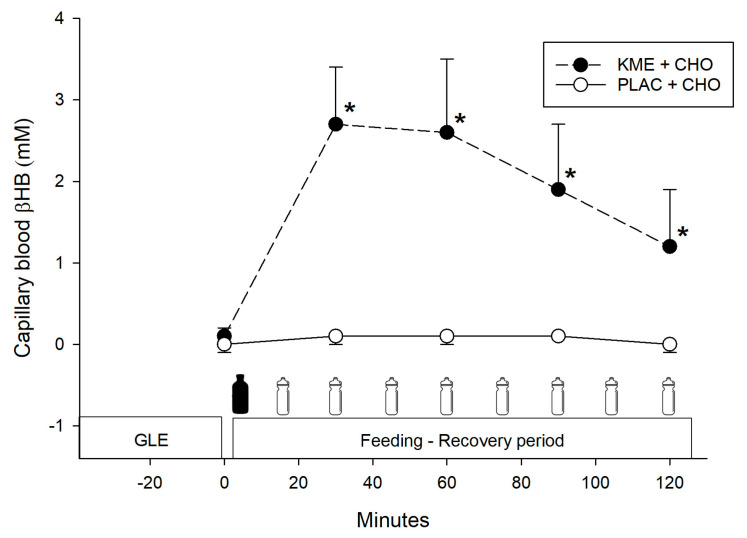
Capillary blood βHB concentration. Values are means ± SD. GLE = glycogen-lowering exercise; KME + CHO = ketone monoester + carbohydrates; PLAC + CHO = iso-energetic placebo + carbohydrates. * Significantly greater (*p* < 0.001) versus PLAC + CHO. 
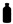
 = treatment ingestion (KME or PLAC). 
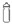
 = CHO drink. βHB = beta-hydroxybutyrate.

**Figure 3 nutrients-16-00932-f003:**
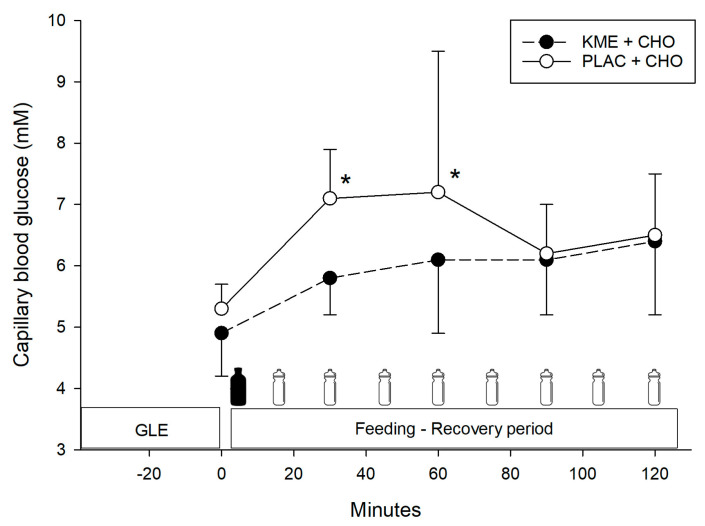
Blood glucose concentration. Values are means ± SD. GLE = glycogen-lowering exercise; KME + CHO = ketone monoester + carbohydrates; PLAC + CHO = iso-energetic placebo + carbohydrates. * Significantly greater (*p* ≤ 0.01) versus PLAC + CHO. 
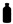
 = treatment ingestion (KME or PLAC). 
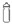
 = CHO drink.

**Figure 4 nutrients-16-00932-f004:**
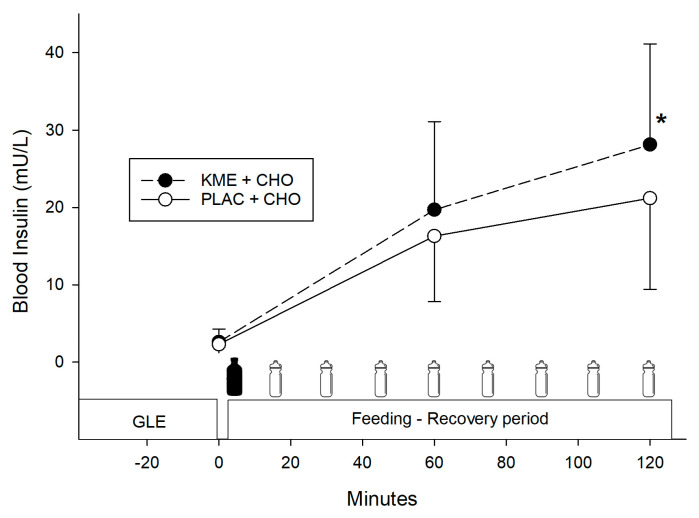
Blood insulin concentration. Values are means ± SD. GLE = glycogen-lowering exercise; KME + CHO = ketone monoester + carbohydrates; PLAC + CHO = iso-energetic placebo + carbohydrates. * Significantly greater (*p* = 0.002) versus PLAC + CHO. 
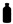
 = treatment ingestion (KME or PLAC). 
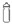
 = CHO drink.

**Figure 5 nutrients-16-00932-f005:**
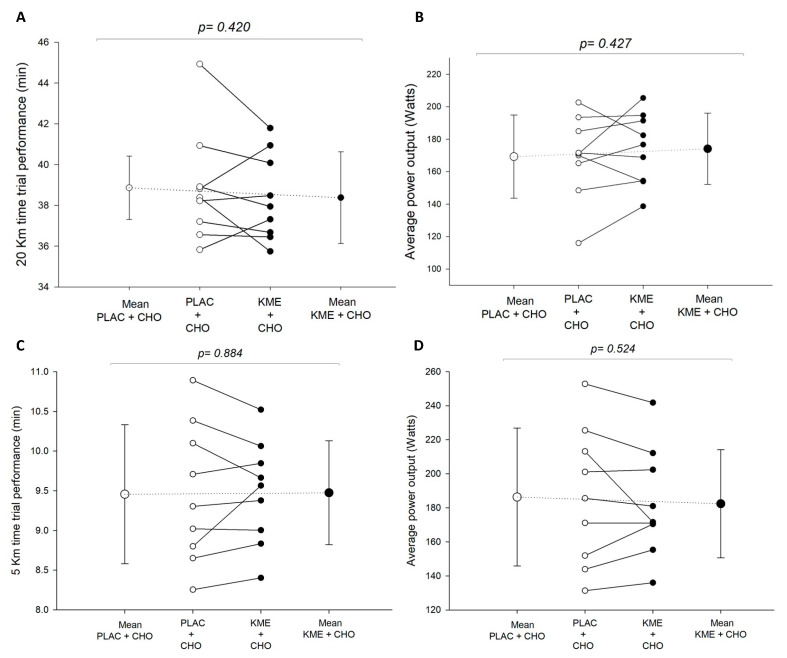
Group means (dotted line) and individual scores (solid lines) for (**A**) time to complete and (**B**) average PO during TT_20km_ as well as (**C**) time to complete and (**D**) average PO during TT_5km_. Values are means ± SD. KME + CHO = ketone monoester + carbohydrates (closed circles); PLAC + CHO = iso-energetic placebo + carbohydrates (open circles).

**Table 1 nutrients-16-00932-t001:** Participants’ characteristics and dietary intake (*n* = 9).

Characteristics	Value
Height, cm	175.6 ± 5.3
Body mass, kg	72.9 ± 7.7
Age, y	28.0 ± 5.0
Body fat ^a^, %	12.2 ± 3.2
Wmax ^b^	314.6 ± 42.3
VO_2_max ^c^, mL·kg^−1^·min^−1^	56.2 ± 5.8
Intakes ^d^	
Energy, kcal·kg^−1^·d^−1^	32.2 ± 3.5
Carbohydrate, g·kg^−1^·d^−1^	3.5 ± 0.4
Protein, g·kg^−1^·d^−1^	1.3 ± 0.3
Fat, g·kg^−1^·d^−1^	1.6 ± 0.4

Values are means ± SD, *n* = 9. Wmax, maximum cycle wattage; VO_2_max, maximum oxygen consumption. ^a^ Body fat was measured using a Bod Pod^®^ (COSMED, Rome, Italy). ^b^ Maximum wattage was measured using a ramp incremental test on a Velotron bicycle (Racer Mate, Seattle, WA, USA). ^c^ Maximum oxygen consumption was measured using breath-by-breath gas analysis during the incremental cycle max test (Vmax Legacy, SensorMedics, Yorba Linda, CA, USA). ^d^ Energy intake and macronutrient breakdown were based on a 2 d dietary record analysis.

## Data Availability

The authors will make the deidentified raw data set available upon reasonable requests.

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
