# Peer review of "Ketone Monoester Followed by Carbohydrate Ingestion after Glycogen-Lowering Exercise Does Not Improve Subsequent Endurance Cycle Time Trial Performance"

_nutrients, 2024, doi:10.3390/nu16070932_

Round 1
Reviewer 1 Report
Comments and Suggestions for Authors
I read the manuscript “Ketone ester plus carbohydrate supplementation following glycogen lowering exercise does not improve subsequent 20 km cycle time trial performance” with great interest. Overall, the manuscript has several strengths, including the randomized placebo controlled cross over design and the performance time trials. However, there are also several components which could be improved, including sample size calculation, reporting effect sizes, and a limitation section in the discussion.
Abstract:
Perhaps put the dose that was used during recovery for both the ketone and the carbohydrates.
Is a lower blood glucose concentration due to greater carbohydrate uptake into the muscle. This may also be explained by the greater insulin.
Introduction:
“enhancing protein synthesis”. I am unclear how altering protein synthesis would influence glycogen re-synthesis or is this a different mechanisms? Was the activation of the mTORC1 due to the protein or the ketones? Again what is the mechanism(s)?
Please state your hypothesis.
Why was a 20km and 5 km time trial selected?
Can you describe how randomization was done? Did you use computer software?
Study design: It is a bit award when day is abbreviated by “d” in a middle of a sentence, just for clarity please write out day and weeks unless used in units. Please check the entire manuscript.
Isn’t every time trial a “best effort” time trial? This terminology is odd to me.
It should be more clear the dose of the KME (Line 153).
Line 162: TT should be “TTs”
What was the rationale for a 20 km time trial followed by a 5 minute rest then a 5 km time trial?
Line 170: is it called a Tukey’s Honest Significant Difference or just Honest Significant difference?
Results:
3.1. If there is a significant interaction effect, it is not necessary or appropriate to examine the main effects.
Again line 192-193. It is unnecessary to report the main effects since there is a maa significant interaction effect similarly, Lines 200-201.
Figure 4: Lines 209-210 figure legend has some errors. Please check it closely.
Lines 217: there is “.” That needs to be deleted.
Figure 5 is great. I like the individual data and group means and SD. However, the quality of the figure needs to be improved. Is it possible to make it more clear?
It may be worth reporting effect sizes.
Was a priori sample size performed?
Was any metabolic data collected during the time trials, this would have allowed you to determine RER and estimate carbohydrate and fat oxidation? Is it possible that ketone could have enhanced glycogen re-synthesis but may impair glycolysis during exercise and thus not improve performance.
Line 308: There is an extra “.”
The discussion could use a limitation section. Example, no measure of muscle glycogen. Study only done in males. Small sample size. Etc.
Comments on the Quality of English Language
The written was very clear. There were a couple grammatical edits that could be improved and were indicated in the comments provided.
Reviewer 2 Report
Comments and Suggestions for Authors
The manuscript by Quinones et al. aims to determine whether supplementation with ketone esters + carbohydrates is more efficient than carbohydrate supplementation alone during the recovery period of a glycogen lowering exercise on subsequent exercise performance. The data presented are of interest and provide new information in this field of research. However, there are significant shortcomings in the current methodology and interpretation of data which dampened my enthusiasm for the publication of the data presented in its current form.
General Comments
· The title is very informative, but had to be read more than once to understand. Making it more concise or organizing the events in a different sequence could help. The results from the 5km time trial should also be mentioned (in addition to the 20km trial).
· The figures appear blurry (especially number 1 and 5).
· The methods state that the participants get one ketone drink, then several drinks with carbohydrates. Often, the text is written in a way that suggests the participants receive ketones and carbs simultaneously (i.e., in the same drink). A couple of instances are on lines 12 and 87 – could change the wording to “supplementation with exogenous ketones followed by carbohydrates”, as an example.
Abstract
· I would suggest removing the numbers in the brackets to make the sentences easier to follow.
· I would recommend not defining acronyms in the abstract, unless they pertain to the treatments (CHO, KME).
· Line 12: I would recommend modifying the wording of this sentence, to clearly state the objective of the study. For example, “The objective of this study was to test the efficacy of ketone monoester…”.
· Lines 20-22: Should clarify what the times mean (e.g., minutes after receiving the first beverage).
Introduction
· Since protein was not used in the study, I would suggest removing the acronym to facilitate reading. Also, the effect of protein can be better integrated into the text, by making comparisons with other nutrients.
· Lines 44-46: I think that ketone bodies need to be introduced in more detail. βHB needs to be introduced, and what tissues use ketone bodies could also be explained. Also, references that support the statement that ketone bodies are an important source of energy during exercise should be cited (reference 9 only refers to their role as signaling molecules).
· Line 46: How are ketone bodies “critical signaling metabolites” in this context? To my knowledge little is known about how ketone bodies may signal during exercise, recovery, etc. βHB is known to have signaling effects but this statement needs to be more specific.
· Line 55 onwards: When looking at the references, it seems like all the studies gave the ketone supplement after exercise (recovery period). More information should be provided on these studies in order to make this clear.
Methods
· Based on Figure 1, it seems as if the 5 km time trial was conduced after the 20 km trial. Was there a reason this order was chosen? Why was it decided to conduct the 5 km time trial following the 20 km time trial? A discussion point could be whether the results could have been different if the order was reversed. Furthermore, the order of the time trials should be clarified in the methods.
· The protocol to deplete muscle glycogen content did probably decrease glycogen content, however, glycogen content was not measured in this study to double check that it was the case. Furthermore, the cited manuscript (8) didn't measure glycogen content with a similar protocol, and the other reference cited (18) is a review. Therefore, there is no proof that this exercise protocol really depleted glycogen content. This is a limitation of the study that should be acknowledged more deeply, unless muscle biopsies were performed and the authors can measure glycogen content pre and post exercise.
· Why were the blood draws stopped after 2h (i.e., why was a sample not taken at the end of the 4h recovery period)? It would have been informative to confirm whether blood glucose and insulin were different prior to beginning the time trials.
· Section 2.2 could be condensed with the sections below it – most of the content in section 2.2 is subsequently repeated in more detail.
· Line 72: I think the research ethics approval number should be provided.
· Would suggest moving lines 153-155 with lines 138-139, so that all information on the recovery drinks is together.
Results
· The captions for Figure 3 and 4 are missing the bottle images before the equal signs.
· Ensure that all figures have an in-text reference.
· The p values in brackets to make the text harder to read, so would suggest removing them from the text.
· Figure 5: The overall format of the graph was well chosen. However, moving the KME group to the right and moving the PLAC to the left (i.e., reversing the current format) would make the results easier to interpret.
Discussion
· Lines 225-229: The sentence is too long, so should be condensed or split in two.
· Line 232: Specify that insulin is greater only at 2 hours.
· Lines 235-236: I don’t think you can assume that glycogen resynthesis was similar between the groups based on the data presented. Glycogen levels would have to be measured to determine this.
· Lines 249-250: It is mentioned that βHB activates the insulin signaling pathway. However, only one reference (#15) showed this effect of βHB, and this was shown in isolated muscles from mice treated with 4 mM βHB ex vivo for 15 min, which is very different from the present study. Further, the same effect was not observed after 2 hours. Ref #12 showed that KE supplementation in humans doesn't increase the activation of the insulin pathway.
· Lines 298-300: Since muscle glycogen levels and hepatic glucose production were not measured, it should be clear that this remains speculative.
· Overall, the discussion feels like a summary of previous data with not enough explicit interpretation of the data. Reasons why glucose and insulin levels could be different, specifically in this study, should be discussed. Also, factors besides muscle glycogen that can influence exercise performance could be considered.
· A paragraph clearly stating the limitations of the study should be included. Lines 312-314 seem like limitations, but should be stated more clearly.
Reviewer 3 Report
Comments and Suggestions for Authors
Thank you for the opportunity to review this manuscript which investigated the effect of ketone monoester (KME) plus carbohydrate supplementation during recovery from glycogen-depleting exercise on subsequent time trial performance. The main finding of the study was that KME plus carbohydrate ingestion during recovery did not enhance subsequent time trial performance compared to ingestion of an iso-energetic placebo plus carbohydrate. While the results of this study are interesting and stimulate further research questions, there are several limitations of the study which require clarification.
Why was CHO only provided during the first two hours of recovery? This needs to be justified in the manuscript. Why not provide CHO for the full 4 hours of recovery?
What was the rationale for designing the study to investigate 20km TT performance, followed almost immediately by 5km TT performance? Again, this needs to be outlined and justified in the manuscript. As far as I am aware this isn’t a usual pattern of performance testing that an athlete may undergo.
It isn’t clearly stated how the placebo+CHO was energy-matched to the KME+CHO trial. In line it states that details regarding the placebo are below, but I can’t find this information in the methods section. Please state how an isocaloric placebo was achieved and what impact that had on the total quantity of carbohydrate consumed in the placebo vs. KME+CHO trial.
The paragraph starting on line 237 discusses the potential impact of KME supplementation on glycogen resynthesis. Given the lower glucose concentrations and greater insulin response in the KME+CHO trial, is it possible that KME+CHO increased oxidative glucose disposal rather than increasing glycogen resynthesis?
Related to the differences in blood glucose response between trials, was the lower blood glucose response in the KME+CHO trial simply related to the lower overall CHO intake observed in this trial?
Round 2
Reviewer 2 Report
Comments and Suggestions for Authors
Thank you for addressing my comments. Please see the attachment.

Author Response
Our response and our revised manuscript are enclosed

Reviewer 3 Report
Comments and Suggestions for Authors
All my comments/feedback have been addressed. Thank you.
Author Response
No revisions were suggested.
